# The Research of Antagonistic Endophytic Bacterium *Bacillus velezensis* CSUFT-BV4 for Growth Promotion and Induction of Resistance to Anthracnose in *Camellia oleifera*

**DOI:** 10.3390/microorganisms12040763

**Published:** 2024-04-10

**Authors:** Yuan He, Xinyu Miao, Yandong Xia, Xingzhou Chen, Junang Liu, Guoying Zhou

**Affiliations:** 1Key Laboratory of National Forestry and Grassland Administration on Control of Artificial Forest Diseases and Pests in South China, Central South University of Forestry and Technology, Changsha 410004, China; jassyyouz@163.com (Y.H.); mxy01312000@163.com (X.M.); xiaxyd@126.com (Y.X.); cxz273711587@163.com (X.C.); 2Hunan Provincial Key Laboratory for Control of Forest Diseases and Pests, Central South University of Forestry and Technology, Changsha 410004, China; 3Key Laboratory of Cultivation and Protection for Non-Wood Forest Trees, Central South University of Forestry and Technology, Changsha 410004, China

**Keywords:** *C. oleifera* anthracnose, antagonistic endophytic bacterium, growth-promoting, inducing immunity

## Abstract

*Camellia oleifera* (*C. oleifera*) is one of the four main, woody, edible oil tree species in the world, while *C. oleifera* anthracnose is mainly caused by the fungus *Colletotrichum fructicola* (*C. fructicola*), which severely affects the yield of *C. oleifera* and the quality of tea oil. *Bacillus velezensis* (*B. velezensis*) CSUFT-BV4 is an antagonistic endophytic bacterium isolated from healthy *C. oleifera* leaves. This study aimed to investigate the biocontrol potential of strain CSUFT-BV4 against *C. oleifera* anthracnose and its possible functional mechanism, and to determine its growth-promoting characteristics in host plants. In vitro, CSUFT-BV4 was shown to have efficient biofilm formation ability, as well as significant functions in the synthesis of metabolic substances and the secretion of probiotic substances. In addition, the CSUFT-BV4 fermentation broth also presented efficient antagonistic activities against five major *C. oleifera* anthracnose pathogens, including *C. fructicola*, *C. gloeosporioides*, *C. siamense*, *C. camelliae*, and *C. kahawae*, and the inhibition rate was up to 73.2%. In vivo, it demonstrated that the growth of *C. oleifera* treated with CSUFT-BV4 fermentation broth was increased in terms of stem width, plant height, and maximum leaf area, while the activities of various defense enzymes, e.g., superoxide dismutase (SOD), phenylalanine aminotransferase (PAL), and polyphenol oxidase (PPO), were effectively increased. The remarkable antagonistic activities against *C. oleifera* anthracnose, the growth-promoting characteristics, and the induction of host defense responses indicate that endophytic bacterium CSUFT-BV4 can be effectively used in the biological control of *C. oleifera* anthracnose in the future, which will have a positive impact on the development of the *C. oleifera* industry.

## 1. Introduction

*Camellia oleifera* (*C. oleifera*), as an important, woody, edible oil tree species in the world, is also unique to China, and its tea oil contains high levels of unsaturated fatty acids, known as “oriental olive oil” [1]. The cultivated area of *C. oleifera* in China is rapidly increasing due to its excellent economic and nutritional value, and yet *C. oleifera* diseases are also becoming more serious. *C. oleifera* anthracnose is one of the major diseases of *C. oleifera*, commonly caused by the fungus *Colletotrichum fructicola* (*C. fructicola*), which can lead to flower and fruit drops, together with large losses in *C. oleifera* quality and tea oil production [2]. Generally, *C. oleifera* anthracnose can cause about 20 to 40% of damage to *C. oleifera* seeds, and the oil production of diseased *C. oleifera* is only 50% that of healthy *C. oleifera*, or even less [3].

The pathogens that cause *C. oleifera* anthracnose are various and involve several species of the genus Anthrax. The traditional classification and identification of the genus Anthrax are mainly based on morphological and cultural characteristics; however, the morphological characteristics of colonies, conidia, and appressoria are similar among the species of the genus Anthrax, making accurate identification difficult [4]. At present, it is mainly identified by the morphological identification of strains and multi-gene sequence information. The most common *C. oleifera* anthracnose pathogens include *C. fructicola*, *C. gloeosporioides*, *C. siamense*, *C. camelliae*, and *C. kahawae*, of which the most widely distributed pathogen is *C. fructicola* [5]. Nowadays, the means of controlling *C. oleifera* anthracnose are still mainly through chemical control, which can quickly resist the attack of pathogens. The long-term use of chemicals has caused the pathogens of *C. oleifera* anthracnose to adapt to common agents, increasing resistance [6]. In addition, the massive use of chemicals will not only cause significant pollution to the ecological environment, but also lead to pesticide residues in *C. oleifera* fruits, endangering human health, affecting the quality of tea oil, and influencing the health development of *C. oleifera* industry [7]. In recent years, China has been intensively pursuing environmental protection policies and paying more attention to environmental protection, pollution control, and ecological restoration [8]. With the growing awareness of environmental protection, the use of healthier means to control plant diseases is receiving more attention. Biological control has gradually become an effective alternative to chemical control due to its environmentally friendly, safe, and efficient characteristics, and the application and research of biological control agents have made significant progress in several aspects [9,10]. Endophytic bacteria can effectively colonize host plants and co-evolve without causing substantial damage [11]. Plant endophytes are vital components of the plant micro-ecosystem and are more effective in controlling pathogenic infections than microorganisms that are active outside the plant without causing visible disease [12]. The biological control mechanisms of endophytic bacteria include nutrient competition, the induction of systemic resistance in plants, and the secretion of metabolic substances that can properly promote plant growth, stimulate the host defense responses, and resist diseases and stress. Therefore, they have received widespread attention from researchers [13]. The screening and effective utilization of antagonistic endophytic bacteria are currently potential means to fight against *C. oleifera* anthracnose, which is a new research direction for the biological control in *C. oleifera*.

*Bacillus* spp. are a type of Gram-positive bacteria that can produce massive amounts of resistant endospores. It can reproduce rapidly in large quantities and produce a variety of antimicrobials and enzymes, and it has a broad spectrum of anti-bacterial activity and strong resistance. It is the ideal beneficial microorganism and is widely used for the biological prevention of plant diseases [14]. In addition, endophytic bacteria have excellent growth-promoting characteristics. They promote the uptake of soil minerals and nitrogen and the secretion of plant growth hormones, principally through phosphorus solubilization, potassium solubilization, and nitrogen fixation [15,16]. For example, the biofilm formation of *Bacillus subtilis* 916 significantly affects its colonization in the host plants, which in turn directly affects the biological control of rice sheath blight [17]. In indoor activity tests, six *Bacillus cereus* strains isolated by Song with broad-spectrum inhibitory activity were found to effectively inhibit the mycelial growth of several pathogens, including *Sclerotium rolfsii*, *Botrytis cinerea,* and *Sclerotinia sclerotiorum* [18]. *B. velezensis* LM2303 can induce systemic resistance through the production of antimicrobial substances and promote plant growth and its nutrient competition, effectively controlling Fusarium head blight [19]. The three strains of *Bacillus* spp. isolated from sour berry fruits by Zhang can produce protease and cellulase, which effectively inhibit the mycelial normal growth of the *C. oleifera* anthracnose pathogen [20]. In addition, *B. velezensis* HM3-3 effectively promoted the growth of soybean and increased the activities of defense enzymes such as catalase (CAT), polyphenol oxidase (PPO), and superoxide dismutase (SOD) in soybeans after spraying with the HM3-3 fermentation broth [21]. Currently, there are relatively few reports on the inhibitory activity of *B. velezensis* against *C. oleifera* anthracnose pathogens and its growth-promoting effect with different *C. oleifera* cultivars, and the mechanism has not been clearly investigated.

Biological control is a promising method for controlling *C. oleifera* anthracnose. The endophytic bacterium *Bacillus velezensis* CSUFT-BV4 was isolated from healthy *C. oleifera* leaves. By the full utilization of CSUFT-BV4-induced *C. oleifera* defense enzyme activities, the growth-promoting characteristics and antagonistic effect on *C. oleifera* anthracnose pathogens can be effective against anthracnose disease. This study aimed to investigate the antagonistic activity of CSUFT-BV4 against *C. oleifera* anthracnose and the potential mechanism of action, and provide a theoretical basis for the biological control of *C. oleifera* anthracnose.

## 2. Materials and Methods

### 2.1. Test Strains and Plants

The pathogenic fungi *C. fructicola*, *C. gloeosporioides*, *C. siamense*, *C. camelliae,* and *C. kahawae* were isolated from the diseased leaves of *C. oleifera* in an early stage and were provided by the Hunan Provincial Key Laboratory for Control of Forest Diseases and Pests (Changsha, China). The endophytic bacterium CSUFT-BV4 was isolated from the leaves of healthy *C. oleifera* in Youxian County, Zhuzhou City, Hunan, China, and was preserved by the Key Laboratory of National Forestry and Grassland Administration on Control of Artificial Forest Diseases and Pests in South China (Changsha, China). The three-year-old *C. oleifera* cultivars “Huashuo” and “Huajin” were purchased from Hunan Tianhua *C. oleifera* Technology Co., Ltd., Youxian Country of Zhuzhou, Hunan, China (26°52′5″ N, 113°19′25″ E).

### 2.2. Molecular Biology Identification of CSUFT-BV4

The strain CSUFT-BV4 was inoculated into Luria Bertani (LB) liquid medium and incubated for 24 h at 37 °C with continuous shaking. Genomic DNA was extracted from bacterial cultures using the TIANamp Bacteria DNA Kit. For the amplification of the bacterial 16SrDNA gene sequences, the extracted genomic DNA served as the template in a polymerase chain reaction (PCR). The PCR reaction utilized the bacterial universal primers 27F (5′-AGAGTTTGATCCTGGCTCAG-3′) and 1492R (5′-GGTTACCTTGTTACGACTT-3′). The PCR running conditions consisted of an initial denaturation step at 94 °C for 30 s, followed by 34 cycles of denaturation at 98 °C for 10 s, annealing at 55 °C for 30 s, and then extension at 72 °C for 1 min, and one final extension at 72 °C for 2 min. The PCR products were confirmed by electrophoresis on 1% agarose gel to visualize the amplified fragments. The products were then sent to Shanghai Biotech Company for sequencing. The highly homologous gene sequences of the 16SrDNA gene were selected from the NCBI website (https://www.ncbi.nlm.nih.gov/, accessed on 1 March 2024) as reference sequences for further analysis. Sequences analysis of the cloned fragments was performed using MEGA version 11.0.13 software, which facilitates alignments and the editing of multiple sequences. Finally, a phylogenetic tree was constructed using the neighbor-joining method.

### 2.3. Determination of the Biofilm Formation Ability of CSUFT-BV4

The bacterial suspensions were prepared by inoculating single colonies of endophytic bacterium CSUFT-BV4 into 50 mL of liquid LB and incubating for 12 h at 28 °C and 180 rpm. Two milliliters of the suspensions was inoculated into 100 mL of LB liquid medium and incubated at a constant temperature of 28 °C for 24 h to prepare a fermentation broth of CSUFT-BV4. The fermentation broth was diluted with LB liquid to reach a concentration of 10^8^ CFU/mL. An amount of 1 mL of the above-diluted CSUFT-BV4 fermentation broth was added into a 48-well plate, and the plate was placed in a constant-temperature incubator, incubated for 24 h at 28 °C. The bacterial cultures were transferred to new blank wells and incubated for 24 h, and the real-time monitoring of biofilm formation was performed in the cell culture plate. Sterilized liquid LB was used as a control, and each group was repeated three times. The culture medium and free cells under the biofilm in the 48-well plate were aspirated with a 1 mL sterile syringe. An amount of 1 mL of 0.1% crystal violet staining solution was added to the 48-well plate, stained for 20 min, and washed with sterile water for 10 times. After all the crystal violet staining solution was washed off, the biofilm stained by crystal violet was dissolved with a buffer solution of 80% ethanol and 20% acetone. Finally, the 1 mL aspirated 10-fold dilution of the mixed solution was transferred into a cuvette and measured at 570 nm with a UV spectrophotometer [22].

### 2.4. Determination of the Metabolites and Growth Promotion Characteristics of CSUFT-BV4

The production of various lytic enzymes and bioactive metabolites is an important mechanism for many antibiotic bacteria [23]. CSUFT-BV4 was cultured as described in Section 2.3 and diluted to 10^8^ CFU/mL with sterilized LB liquid medium. An amount of 2 μL of the CSUFT-BV4 fermentation broth was pipetted and added dropwise to the three vertices of the protease, amylase, cellulase, and β-1,3-glucanase assay plates in the form of an equilateral “Δ”, and the plates of each group were cultured in a thermostatic incubator, 28 °C. Each group was repeated three times, and the plates were observed to determine whether transparent circles appeared after 2–4 days [24,25].

Following the method described above, 2 μL of the CSUFT-BV4 fermentation broth at the concentration of 10^8^ CFU/mL was aspirated and inoculated into Pikovskaya organophosphate, inorganic phosphorus, Ashby, and potassium solubilizing assay plates at 28 °C for 5 days to observe whether transparent circles appeared [26,27]. The strain was tested for indoleacetic acid (IAA)-producing ability by Salkowski’s colorimetric method and for siderophore carrier-producing ability by the CAS method [28].

### 2.5. Determination of the Growth-Promoting Effect of CSUFT-BV4 on Different C. oleifera Cultivars

It was found that “Huashuo” *C. oleifera* cultivars had the best growth, highest yield, and most stable yield among the “Hua” series cultivars of *C. oleifera* [29]. “Huajin” *C. oleifera* cultivars had the highest dry kernel weight, the highest dry seed yield, and the highest seed and fruit oil content, as well as the richest stored nitrogen source [30].

Therefore, for the purpose of comparing the effect of CSUFT-BV4 on the growth of different *C. oleifera* cultivars, CSUFT-BV4 was cultured according to the method described in Section 2.3 and diluted to the concentration of 10^8^ CFU/mL with sterilized LB liquid medium. The CSUFT-BV4 fermentation broth was inoculated into the three-year-old “Huashuo” and “Huajin” *C. oleifera* cultivars by using the following inoculation methods—(1) leaf spray; (2) root irrigation; (3) leaf spray + root irrigation, 40 mL per plant—and the CSUFT-BV4 fermentation broth cultivated under the same conditions was replenished every half a month. The stem width, plant height, and maximum leaf area of different *C. oleifera* cultivars were recorded at 45 and 90 days, respectively. Sterilized liquid LB medium was used as a control, and each group was repeated three times.

### 2.6. Antagonistic Activity of CSUFT-BV4 against Pathogens of C. oleifera Anthracnose In Vitro

The endophytic bacterium CSUFT-BV4 was cultured according to the method described in Section 2.3 and diluted to the concentration of 10^8^ CFU/mL with sterilized LB liquid medium. The antagonistic activity of the CSUFT-BV4 fermentation broth against the pathogens of *C. oleifera* anthracnose was tested by the plate stand-off test. Five species of *C. oleifera* anthracnose pathogens, *C. fructicola*, *C. gloeosporioides*, *C. siamense*, *C. camelliae* and *C. kahawae*, all of which were growing strongly on the plates, were cut into 5 mm pieces and inoculated into the center of the PDA plates. An amount of 2 μL of the CSUFT-BV4 fermentation broth was aspirated and inoculated into the three corner points in the form of an equilateral “Δ”. The PDA plate was only inoculated with 5 mm pathogenic clumps as a control at 28 °C for 5 days, and each group was repeated three times. The diameter of the pathogenic colonies was measured by the cross method and the area of the treatment groups was measured by the irregular graph area calculation method to determine the antagonistic activity of CSUFT-BV4 against *C. oleifera* anthracnose in vitro [31].

The formula for inhibition rate is as follows:
Inhibition rate (%) = (control colony area − treatment colony area)/control colony area × 100.

### 2.7. Inhibitory Ability of CSUFT-BV4 on C. oleifera Anthracnose In Vivo

Five species of *C. oleifera* anthracnose pathogens, *C. fructicola*, *C. gloeosporioides*, *C. siamense*, *C. camelliae* and *C. kahawae*, were cut into 5 mm fungi clumps and placed into the PDA liquid medium at 28 °C for 4 days. After the end of the incubation, isolated spores were subjected to 5000 rpm for 10 min. The spore solution was diluted to 10^7^ CFU/mL with sterile water. CSUFT-BV4 was cultured according to the method described in Section 2.3 and diluted to 10^8^ CFU/mL with sterilized LB liquid medium. *C. oleifera* leaves were pricked but not pierced, immersed in a petri dish containing CSUFT-BV4 fermentation broth for 20 min, and then removed and left to air dry. An amount of 5 μL of the diluted spore solution was pipetted into the puncture wound of *C. oleifera* leaves soaked with CSUFT-BV4 fermentation broth and placed in petri dishes with moist sterile cotton at 28 °C for 5 days. Each group was repeated three times. The diameter of the pathogenic colonies was measured by the cross method to determine the inhibitory ability of the endophytic bacterium CSUFT-BV4 against *C. oleifera* anthracnose under isolated leaves. Refer to Section 2.6 for the calculation of the inhibition rate formula.

### 2.8. Variation in Defense Enzyme Activities in C. oleifera Leaves after Different Treatments

The CSUFT-BV4 fermentation broth was cultured according to the method described in Section 2.3 and diluted to 10^8^ CFU/mL with sterilized liquid LB medium. The pathogen *C. fructicola* spore solution was prepared according to the method described in Section 2.7 and sprayed on the leaves of *C. oleifera* at 40 mL per plant, respectively. Four treatment groups were established: 1 (CK): sprayed with sterilized LB liquid medium alone; 2 (*C. fructicola*): inoculated with the *C. fructicola* spore solution alone; 3 (CSUFT-BV4): sprayed with the CSUFT-BV4 fermentation broth alone; 4 (*C. fructicola* + CSUFT-BV4): both inoculated with the *C. fructicola* spore solution and the CSUFT-BV4 fermentation broth. Each treatment group was repeated three times. The levels of defense enzyme activities of *C. oleifera* leaves were determined at nine time points (0, 1, 2, 3, 4, 5, 6, 7, and 14 days).

The samples of 0.2 g of *C. oleifera* leaves and 5 mL of phosphate buffer (0.05 mol/L pH 6.8) were homogenized in an ice bath and centrifuged at 4 °C and 12,000 rpm for 10 min, and the supernatant was retained as the crude enzyme extract. The activities of superoxide dismutase (SOD), polyphenol oxidase (PPO), and phenylalanine ammonia-lyase (PAL) were determined according to the instructions of the assay kits (Gerace Biotechnology Co., Ltd., Suzhou, China).

### 2.9. Data Analysis

Microsoft Excel 2019 and SPSS v.26.0 software were utilized for data processing and statistical analysis in this study. Duncan’s new multiple range test was used to detect significant differences between groups (*p* < 0.05), expressed as mean ± standard deviation.

## 3. Results

### 3.1. Molecular Biological Identification of CSUFT-BV4

The amplified 16S rDNA gene fragment of CSUFT-BV4 was sequenced to obtain a full-length sequence of 1239 bp. The 16S rDNA gene sequences of CSUFT-BV4 were aligned and a phylogenetic analysis was conducted. The phylogenetic analysis revealed that endophytic bacterium CSUFT-BV4 clustered together with *Bacillus velezensis*, with a branch support value of 99.00% (Figure 1). Therefore, strain CSUFT-BV4 could be identified as *Bacillus velezensis*.

### 3.2. The Biofilm Formation Ability of CSUFT-BV4

Plates were left at 28 °C for 48 h, the liquid LB remained clarified, and the surface was transparent, without any bacterial film formation observed (Figure 2A-a). After incubation for 24 h, there was an accumulation of bacteria on the surface, accompanied by the formation of yellow striped bacterial film aggregates (Figure 2A-b). After incubation for 48 h, a dense and tight yellow bacterial membrane was formed on the surface and folds appeared (Figure 2A-c). This indicated that CSUFT-BV4 has the ability of biofilm formation. According to the quantitative results (Figure 2B), the OD_570_ value after incubation for 48 h was 3.996, which was significantly higher than the amount of biofilm after incubation for 24 h.

### 3.3. The Metabolic Substance Synthesized and Growth-Promoting Characteristics of CSUFT-BV4

The metabolites synthesized by the endophytic bacterium CSUFT-BV4 were measured and the results are shown in Table 1. After incubation for 3 days, CSUFT-BV4 grew well on protease, amylase, cellulase, and β-1,3-glucanase testing plates, with clear and obvious hyaline circles around the colonies that gradually expanded with incubation time, suggesting that CSUFT-BV4 was able to stably produce protease (Figure 3A), amylase (Figure 3B), cellulase (Figure 3C), and β-1,3-glucanase (Figure 3D). The production of these metabolites may be important for CSUFT-BV4 to help *C. oleifera* resist plant diseases and inhibit the growth of pathogens.

The growth-promoting characteristics of CSUFT-BV4 are shown in Table 2. The ability to produce IAA was detected by Salkowski’s colorimetric assay, the mixture treated with the addition of CSUFT-BV4 had a light pink color (Figure 4A-b), and the control without the bacteria had a light-yellow color (Figure 4A-a); therefore, strain CSUFT-BV4 possesses the ability to produce IAA. In the Ashby texting plate, CSUFT-BV4 was able to grow normally in the plate and form moist, wrinkled yellowish-white colonies (Figure 4B), showing that CSUFT-BV4 had the ability to fix nitrogen. In the Pikovskaya organic phosphorus activity testing plate, a transparent circle appeared around the colonies of CSUFT-BV4, with a small diameter (Figure 4C). In the PKO inorganic phosphorus activity testing plate, a clear transparent circle appeared around the colonies of CSUFT-BV4 after bromophenol blue staining (Figure 4D). The endophytic bacterium CSUFT-BV4 was able to solubilize inorganic phosphorus better than organic phosphorus. However, there were no transparent circles observed on the potassium and siderophore testing plates, and it was concluded that CSUFT-BV4 was unable to solubilize potassium and secrete siderophore.

### 3.4. The Growth Promotion of “Huashuo” C. oleifera Cultivars under Different Inoculation Methods of CSUFT-BV4

#### 3.4.1. The Effect of Leaf Spray Method on the Growth of “Huashuo” *C. oleifera* Cultivars

In the greenhouse experiment, statistical analysis showed significant differences in stem width between 0, 45, and 90 days in “Huashuo” *C. oleifera* cultivars (Figure 5A). The original stem widths of *C. oleifera* in the LB medium control and 10^8^ CFU/mL CSUFT-BV4 fermentation broth treatment groups were 4 mm and 4.8 mm, respectively, and after 90 days, the values were 4.5 mm and 7 mm, respectively. The stem width of plants treated with CSUFT-BV4 fermentation broth increased by 1.7 mm compared to the control groups. The original plant height values of *C. oleifera* in the LB medium control and 10^8^ CFU/mL CSUFT-BV4 fermentation broth treatment groups were 55.1 cm and 64.9 cm, respectively, and after 90 days, the heights were 63.5 cm and 83.1 cm, respectively. There were significant differences in plant height between 0, 45, and 90 days in “Huashuo” *C. oleifera* cultivars (Figure 5B). The increase in plant height after treatment with CSUFT-BV4 fermentation broth was 18.2 cm, which was 2.17 times higher than the increase in plant height in the control groups. Statistical analysis showed no significant differences in maximum leaf area between 0, 45, and 90 days in “Huashuo” *C. oleifera* cultivars (Figure 5C). The original maximum leaf areas of the test *C. oleifera* in the LB medium control and 10^8^ CFU/mL CSUFT-BV4 fermentation broth treatment groups were 23.01 cm^2^ and 22.15 cm^2^, respectively, and after 90 days, the values were 23.52 cm^2^ and 24.52 cm^2^, respectively. The maximum leaf area increased by 2.37 cm^2^ in the treated groups and 1.87 cm^2^ compared to the control groups. “Huashuo” *C. oleifera* cultivars were inoculated with 10^8^ CFU/mL of CSUFT-BV4 fermentation broth by leaf spray for 90 days, and the growth characterization of the plants is shown in Figure 6.

The 10^8^ CFU/mL CSUFT-BV4 fermentation broth inoculated by the leaf spray method promoted the stem width, plant height, and maximum leaf area of “Huashuo” *C. oleifera* cultivars compared with the CK(LB) treatment.

#### 3.4.2. The Effect of Root Irrigation Method on the Growth of “Huashuo” *C. oleifera* Cultivars

In the greenhouse experiment, the original stem widths of *C. oleifera* in the LB medium control and 10^8^ CFU/mL CSUFT-BV4 fermentation broth treatment groups were 5.3 mm, and after 90 days, the values were 5.6 mm and 6.8 mm, respectively. The stem width of the treatment groups increased by 1.2 mm compared to the control groups (Figure 7A). Statistical analysis showed significant differences in plant height between 0, 45, and 90 days in “Huashuo” *C. oleifera* cultivars. The original plant height values of *C. oleifera* in the LB medium control and 10^8^ CFU/mL CSUFT-BV4 fermentation broth treatment groups were 58.9 cm and 54.2 cm, respectively, and after 90 days, the heights were 66.8 cm and 69.37 cm, respectively. The plant height in the treatment groups increased by 7.3 cm compared to the control groups (Figure 7B). There were no significant differences in maximum leaf area between 0, 45, and 90 days in “Huashuo” *C. oleifera* cultivars. The original maximum leaf areas of the test *C. oleifera* in the LB medium control and 10^8^ CFU/mL CSUFT-BV4 fermentation broth treatment groups were 23.13 cm^2^ and 20.93 cm^2^, respectively, and after 90 days, the values were 23.62 cm^2^ and 22.59 cm^2^, respectively. The maximum leaf area increased by 1.66 cm^2^ in the treated groups and 1.17 cm^2^ compared to the control groups (Figure 7C). “Huashuo” *C. oleifera* cultivars were inoculated with 10^8^ CFU/mL of CSUFT-BV4 fermentation broth by root irrigation for 90 days, and the growth characterization of the plants is shown in Figure 8.

The 10^8^ CFU/mL CSUFT-BV4 fermentation broth inoculated by the root irrigation method promoted the stem width, plant height, and maximum leaf area of “Huashuo” *C. oleifera* cultivars compared with the CK(LB) treatment.

#### 3.4.3. The Effect of Leaf Spray + Root Irrigation Method on the Growth of “Huashuo” *C. oleifera* Cultivars

In the greenhouse experiment, statistical analysis showed significant differences in stem width between 0, 45, and 90 days in “Huashuo” *C. oleifera* cultivars. The original stem widths of the *C. oleifera* in the LB medium control and 10^8^ CFU/mL CSUFT-BV4 fermentation broth treatment groups were 4.8 mm and 5 mm, and after 90 days, the values were 5.3 mm and 7.1 mm, respectively. The stem width of the treatment groups increased by 1.6 mm compared to the control groups (Figure 9A). There were significant differences in plant height between 0, 45, and 90 days in “Huashuo” *C. oleifera* cultivar; the original plant height values of *C. oleifera* in the LB medium control and 10^8^ CFU/mL CSUFT-BV4 fermentation broth treatment groups were 54.1 cm and 57.1 cm, respectively, and after 90 days, the heights were 63.7 cm and 77.07 cm, respectively. The plant height in the treatment groups increased by 10.37 cm compared to the control groups (Figure 9B). There were no significant differences in maximum leaf area between 0, 45, and 90 days in “Huashuo” *C. oleifera* cultivars. The original maximum leaf areas of the test *C. oleifera* in the LB medium control and 10^8^ CFU/mL CSUFT-BV4 fermentation broth treatment groups were 16.87 cm^2^ and 20.86 cm^2^, respectively, and after 90 days, the values were 17.27 cm^2^ and 22.71 cm^2^, respectively. The maximum leaf area increased by 1.85 cm^2^ in the treated groups and 1.45 cm^2^ compared to the control groups (Figure 9C). “Huashuo” *C. oleifera* cultivars were inoculated with 10^8^ CFU/mL of CSUFT-BV4 fermentation broth by leaf spray + root irrigation for 90 days, and the growth characterization of the plants is shown in Figure 10.

The 10^8^ CFU/mL CSUFT-BV4 fermentation broth inoculated by the leaf sprat + root irrigation method promoted the stem width, plant height, and maximum leaf area of “Huashuo” *C. oleifera* cultivars compared with the CK(LB) treatment.

### 3.5. The Growth Promotion of “Huajin” C. oleifera Cultivars under Different Inoculation Methods of CSUFT-BV4

#### 3.5.1. The Effect of Leaf Spray Method on the Growth of “Huajin” *C. oleifera* Cultivars

In the greenhouse experiment, statistical analysis showed significant differences in stem width between 0, 45, and 90 days in “Huajin” *C. oleifera* cultivars, and after 90 days, the stem widths of “Huajin” *C. oleifera* increased by 0.6 mm and 2 mm in the LB medium control and 10^8^ CFU/mL CSUFT-BV4 fermentation broth treatment groups, respectively, and by 3.3 times compared with the CK (LB) groups (Figure 11A). There were significant differences in plant height of “Huajin” *C. oleifera* cultivars treated with 10^8^ CFU/mL fermentation broth between 0, 45, and 90 days (Figure 11B). After 90 days, the plant heights of “Huajin” *C. oleifera* increased by 7.3 cm and 17.7 cm in the LB medium control and 10^8^ CFU/mL CSUFT-BV4 fermentation broth treatment groups, respectively, and by 2.4 times compared with the CK (LB) groups. Statistical analysis showed no significant differences in maximum leaf area between 0, 45, and 90 days in “Huajin” *C. oleifera* cultivars (Figure 11C). The original maximum leaf area of the test *C. oleifera* in the LB medium control and 10^8^ CFU/mL CSUFT-BV4 fermentation broth treatment groups were 18.01 cm^2^ and 20.3 cm^2^, respectively, and after 90 days, the values were 18.58 cm^2^ and 22.4 cm^2^, respectively. The maximum leaf area increased by 2.1 cm^2^ in the treated groups and 1.53 cm^2^ compared to the control groups. “Huajin” *C. oleifera* cultivars were inoculated with 10^8^ CFU/mL of CSUFT-BV4 fermentation broth by leaf spray for 90 days, and the growth characterization of the plants is shown in Figure 12.

The 10^8^ CFU/mL CSUFT-BV4 fermentation broth inoculated by the leaf spray method promoted the stem width, plant height, and maximum leaf area of “Huajin” *C. oleifera* cultivars compared with the CK(LB) treatment.

#### 3.5.2. The Effect of Root Irrigation Method on the Growth of “Huajin” *C. oleifera* Cultivars

In the greenhouse experiment, statistical analysis showed significant differences in stem width between 0, 45, and 90 days in “Huajin” *C. oleifera* cultivars. The original stem widths of *C. oleifera* in the LB medium control and 10^8^ CFU/mL CSUFT-BV4 fermentation broth treatment groups were 6.4 mm and 5.4 mm, and after 90 days, the values were 6.6 mm and 6.7 mm, respectively. The stem widths of the treatment groups increased by 1.1 mm compared to the control groups (Figure 13A). The original plant height values of *C. oleifera* in the LB medium control and 10^8^ CFU/mL CSUFT-BV4 fermentation broth treatment groups were 59.4 cm and 51.6 cm, respectively, and after 90 days, the heights were 63.8 cm and 63.3 cm, respectively. The plant heights in the treatment groups increased by 7.3 cm compared to the control groups (Figure 13B). There were no significant differences in maximum leaf area between 0, 45, and 90 days in “Huajin” *C. oleifera* cultivars. The original maximum leaf areas of the test *C. oleifera* in the LB medium control and 10^8^ CFU/mL CSUFT-BV4 fermentation broth treatment groups were 18.7 cm^2^ and 19.7 cm^2^, respectively, and after 90 days, the values were 19.2 cm^2^ and 21.1 cm^2^, respectively. The maximum leaf area increased by 1.4 cm^2^ in the treated groups and 0.9 cm^2^ compared to the control groups (Figure 13C). “Huajin” *C. oleifera* cultivars were inoculated with 10^8^ CFU/mL of CSUFT-BV4 fermentation broth by root irrigation for 90 days, and the growth characterization of the plants is shown in Figure 14.

The 10^8^ CFU/mL CSUFT-BV4 fermentation broth inoculated by the root irrigation method promoted the stem width, plant height, and maximum leaf area of “Huajin” *C. oleifera* cultivars compared with the CK(LB) treatment.

#### 3.5.3. The Effect of Leaf Spraying + Root Irrigation Method on the Growth of “Huajin” *C. oleifera* Cultivars

In the greenhouse experiment, statistical analysis showed significant differences in stem width between 0, 45, and 90 days in “Huajin” *C. oleifera* cultivars. After 90 days, the stem widths of “Huajin” *C. oleifera* increased by 0.5 mm and 1.6 mm in the LB medium control and 10^8^ CFU/mL CSUFT-BV4 fermentation broth treatment groups, respectively, and by 3.2 times compared with the CK (LB) groups (Figure 15A). There were significant differences in the plant height of “Huajin” *C. oleifera* cultivars between 0, 45, and 90 days (Figure 15B). After 90 days, the plant height of “Huajin” *C. oleifera* increased by 8.8 cm and 17.1 cm in the LB medium control and 10^8^ CFU/mL CSUFT-BV4 fermentation broth treatment groups, respectively, and by 1.9 times compared with the CK (LB) groups. The original maximum leaf area of the test *C. oleifera* in the LB medium control and 10^8^ CFU/mL CSUFT-BV4 fermentation broth treatment groups were 22.6 cm^2^ and 18.03 cm^2^, respectively, and after 90 days, the values were 23.1 cm^2^ and 19.81 cm^2^, respectively. The maximum leaf area increased by 1.78 cm^2^ in the treated groups and 1.28 cm^2^ compared to the control groups (Figure 15C). “Huajin” *C. oleifera* cultivars were inoculated with 10^8^ CFU/mL of CSUFT-BV4 fermentation broth by leaf spray + root irrigation for 90 days, and the growth characterization of the plants is shown in Figure 16.

The 10^8^ CFU/mL CSUFT-BV4 fermentation broth inoculated by the leaf spray + irrigation method promoted the stem width, plant height, and maximum leaf area of “Huajin” *C. oleifera* cultivars compared with the CK(LB) treatment.

### 3.6. Antagonistic Activities of CSUFT-BV4 against C. oleifera Anthracnose In Vitro

The results of the plate stand-off test showed that the fermentation broth of the endophytic bacterium CSUFT-BV4 had a significant inhibitory effect on five pathogen species of *C. oleifera* anthracnose (Figure 17). A clear and distinct transparent inhibitory band appeared around the colonies between CSUFT-BV4 and the five pathogens. The mycelia of the pathogen colonies on the side facing the CSUFT-BV4 colonies were significantly deepened in color, indicated the strongest antagonistic activity of CSUFT-BV4 against *C. fructicola* with an inhibition rate of up to 73.2%, while the antagonistic activity against *C. siamense* was lowest. The inhibition rate still reached 68.8% (Table 3). This showed that the endophytic bacterium CSUFT-BV4 can particularly inhibit the mycelial growth of the *C. oleifera* anthracnose pathogens.

### 3.7. Antagonistic Activities of CSUFT-BV4 against C. oleifera Anthracnose In Vivo

Under the condition of isolated leaves, the inhibitory ability of CSUFT-BV4 is shown in Figure 18. After incubation for 3 days, five pathogen species of *C. oleifera* anthracnose in the control group were able to infect the leaves and form disease spots. In the treatment group, *C. oleifera* leaves were soaked with CSUFT-BV4 fermentation broth and then infected by the five pathogen species. The size of the disease spots on the leaves in each treatment group varied, and the area of the spots was smaller than that of the control group. The pathogen *C. fructicola* had the smallest spot size with the highest inhibition rate of 61.5% and *C. camelliae* had the lowest inhibition rate of 55.4% (Table 4).

Compared to the in vitro test, the inhibition rate of the CSUFT-BV4 fermentation broth against *C. oleifera* anthracnose pathogens under isolated leaves was lower than that against the pathogens in the plate stand-off test. The inhibitory effect was still clearly visible.

### 3.8. The Effect of CSUFT-BV4 on the Activity Level of C. oleifera Defense Enzymes

When plants are exposed to pathogens or external stress, they activate their own defense responses and alter the activity of defense enzymes. Endophytic bacteria are involved in the interaction between the host plant and the pathogens, through physiological and biochemical reactions and signal transduction, activating changes in the level of defense enzymes and inducing the production of small molecular metabolites, which increases the plant’s resistance to the disease. The defense system of *C. oleifera* was also affected by the endophytic bacterium CSUFT-BV4 and the *C. oleifera* anthracnose pathogen *C. fructicola* (Figure 19).

After inoculation with CSUFT-BV4 and *C. fructicola*, the changes in SOD activity in *C. oleifera* are shown in Figure 19A. SOD activity in the treated groups was higher than that of the control group within 14 days. SOD activity in the control groups initially decreased, increased, and finally stabilized, reaching a peak on the 14th day with a maximum value of 396.4 U/g·min^−1^. When inoculated with the pathogen *C. fructicola* alone, the pathogen infection of *C. oleifera* was successful, and the SOD activity showed an increasing trend and reached a peak on the 5th day with a measured value of 451.85 U/g·min^−1^. When inoculated with endophytic bacterium CSUFT-BV4 alone, SOD activity showed a changing trend of decreasing and then increasing, reaching the peak value on the 5th day, with the highest value of 454.89 U/g·min^−1^. Furthermore, when inoculated with both the endophytic bacterium CSUFT-BV4 and the pathogen *C. fructicola*, the SOD activity of *C. oleifera* increased rapidly and showed a trend of increasing and then decreasing, reaching the peak value among all treatment groups on day 4, with the highest value of 525.06 U/g·min^−1^, which was 1.32 times higher than that of the control group. The results showed that the endophytic bacterium CSUFT-BV4 induced the activity of the *C. oleifera* defense enzyme and increased the SOD activity when *C. oleifera* was infected with the pathogen.

The PAL activity changes of *C. oleifera* leaves are shown in Figure 19B. PAL activity in the control group was lower than that in the treated group from day 1 to 5, showing a decreasing and increasing trend, and reaching a peak on day 4 with a maximum value of 215.01 U/·min^−1^. When inoculated with the pathogen *C. fructicola* alone, PAL activity showed an increasing trend and reached a peak on day 6 with a measured value of 242 U/g·min^−1^. When inoculated with endophytic bacterium CSUFT-BV4 alone, the PAL activity of *C. oleifera* increased and then decreased, reaching a peak on day 14 with a maximum value of 251.11 U/g·min^−1^. When inoculated with both the endophytic bacterium CSUFT-BV4 and the pathogen *C. fructicola*, the PAL activity of *C. oleifera* showed a trend of increasing, decreasing, and then increasing, reaching the peak value on the 4th day with the highest value of 311.67 U/g·min^−1^, which was 1.45 times higher than that of the control group. These findings indicated that endophytic bacterium CSUFT-BV4 significantly induced PAL defense enzyme activity in *C. oleifera*.

As shown in Figure 19C, PPO activity in the control group was higher than that of the treatment group on day 0–1, and then PPO activity increased rapidly in the treated group, which was higher than that in the control group. PPO activity in the control group initially increased and then decreased and increased again, reaching a peak on day 4 with a maximum value of 62 U/g·min^−1^. When inoculated with the pathogen *C. fructicola* alone, PPO activity increased and then decreased, reaching its peak on day 4 with a measured value of 83.33 U/g·min^−1^. When inoculated with endophytic bacterium CSUFT-BV4 alone, the PPO activity of *C. oleifera* also increased, decreased, and finally stabilized, reaching a peak of 79.33 U/g·min^−1^ on day 4. When inoculated with both the endophytic bacterium CSUFT-BV4 and the pathogen *C. fructicola*, PPO activity increased slowly from day 0 to 2 and then rapidly increased from day 3, with a general trend of increasing and then decreasing, reaching a peak on the 4th day, with a maximum value of 102.67 U/g·min^−1^, which was 1.67 times higher than that of the control group.

In summary, when *C. oleifera* was infected with the *C. oleifera* anthracnose pathogen, the endophytic bacterium CSUFT-BV4 effectively stimulated the defense response of the host and increased the defense enzymes activities of SOD, PAL, and PPO.

## 4. Discussion

Endophytes that are stably colonized in plants have a stable living space and are not easily influenced by the external environment [32]. In the long-term coevolution process with host plants, endophytes and plants have established a mutually beneficial relationship [33]. Plants can provide nutrients for endophytic microorganisms, while endophytic bacteria can promote the growth and metabolite production of host plants, induce plant resistance, and defend against pathogen invasion [34,35]. As a widely used microorganism to control plant diseases, the biocontrol mechanism of *Bacillus* spp. mainly includes competition for nutrients and spatial locations, the production of antimicrobial substances, lysogeny, the induction of plant defense responses, and the promotion of host plant growth [36,37]. Currently, the main biocontrol *Bacillus* includes *B. velezensis*, *B. subtilis*, *B. amyloliquefaciens,* and *Bacillus thuringiensis*.

The ability to form biofilm is a vital factor in the colonization of hosts by biocontrol strains and can directly affect the effectiveness of host plants to resist the invasion of pathogens [17]. Klein, M. N. et al. found that the stimulation of *Aureobasidium pullulans* ACBL-77 biofilm formation significantly increased host resistance to citrus sour rot disease [38]. Dong found that the wild strain *B. tequilensis* DZY 6715 had a greater biofilm forming ability as well as a better control of *C. oleifera* anthracnose compared to the mutant strain [13]. In this study, the antagonistic endophytic bacterium CSUFT-BV4 possessed a significant biofilm formation ability, which may play an important role in the successful colonization of *C. oleifera* and induce resistance to plant diseases.

*Bacillus* spp. can secrete a variety of metabolic substances which play an important role in breaking down the fungal cell wall and thus resisting fungal diseases [39]. Furthermore, they have the functions of phosphorus dissolution, potassium dissolution, and nitrogen fixation, and they can produce growth-promoting substances, which can effectively promote plant growth. *Bacillus cereus* YN917 isolated by Zhou showed significant antifungal activity against *Magnaporthe oryzae*, and *Bacillus licheniformis* YZCUO202005 isolated by Medison inhibited the growth of five pathogenic fungi and two bacteria [40,41]. The two types of *Bacillus* species have the abilities of secreting various plant-growth-promoting and metabolic substances, such as IAA, siderophore, protease, amylase, cellulase, β-1,3-glucanase, and phosphate solubilizing functions, which can promote host growth and disease resistance under greenhouse conditions. Similarly, in this study, the antagonistic endophytic bacterium CSUFT-BV4 was also able to synthesize a variety of metabolic substances, such as protease, cellulase, amylase, and β-1,3-glucanase, which can directly or indirectly inhibit the growth of pathogenic bacteria and increase the resistance of the plant to disease. In addition, CSUFT-BV4 can secrete a variety of probiotic substances, although it does not have the function of potassium dissolution and siderophore secretion, and it still has the function of IAA production, phosphorus dissolution, and nitrogen fixation, which are directly or indirectly involved in promoting the growth of *C. oleifera*. After the CSUFT-BV4 fermentation broth was sprayed on different *C. oleifera* cultivars, it promoted the growth with an increase in stem width, plant height, and maximum leaf area. The results of the plate stand-off and the isolated leaf test showed that the endophytic bacterium CSUFT-BV4 could effectively inhibit the growth of five species of anthracnose pathogens in *C. oleifera*.

When the plant was attacked by pathogens, the plant body initiated various defense responses based on the stimulus [42]. In plants, systemic acquired resistance (SAR) and induced systemic resistance (ISR) are two important defense mechanisms in response to pathogen infection. SAR is usually mediated by the salicylic acid (SA) signaling pathway, while ISR is mainly mediated by the jasmonic acid (JA) and ethylene (ET) signaling pathways [43]. Cotton plants treated with *B. altitudinis* HNH7 and *B. velezensis* HNH9 showed a significant increase in the expression of SOD, PAL, and PPO activities as well as increased resistance to blight and a significant reduction in the disease condition of cotton [44]. *B. altitudinis* GS-16 isolated from healthy tea leaves by Wu et al. effectively increased the activities of key defense enzymes PPO, SOD and PAL in tea tree, thereby inducing plant resistance and demonstrating strong antagonistic activity against anthracnose in tea tree [45]. Cucumber samples treated with *B. subtilis* YB-04 reported by Xu et al. showed that the host defense response was stimulated with an increase in the activity of PPO, SOD, and PAL, which led to an increase in host resistance to cucumber wilt [46]. Similarly, in this study, SOD, PAL, and PPO activities of *C. oleifera*. were significantly increased by spraying the CSUFT-BV4 fermentation broth during the interaction between *C. oleifera* and the pathogen *C. fructicola*, which induced resistance to anthracnose against anthracnose disease infestation.

The issues proposed for further research include (1) the use of transcriptome sequencing methods to analyze the signal transduction pathways initiated by CSUFT-BV4 treatment; (2) the isolation and identification of additional functional metabolites of CSUFT-BV4, with particular emphasis on their potential to induce systemic resistance in the host; and (3) field trials conducted with CSUFT-BV4, focusing on monitoring the growth promotion and biological control of *C. oleifera* anthracnose disease, to effectively promote the actual production and application in *C. oleifera* industry.

## 5. Conclusions

In this study, the strain CSUFT-BV4, isolated from healthy *C. oleifera* leaves, was functionally tested and found to not only have the ability of biofilm formation, but also have the function of synthesizing a variety of metabolic substances and secreting a series of pro-biotic substances. The endophytic bacterium CSUFT-BV4 fermentation broth had highly significant inhibitory effects against five pathogen species of *C. oleifera* anthracnose in in vivo and in vitro tests and inhibited the growth of *C. oleifera* anthracnose pathogens effectively. When CSUFT-BV4 was inoculated into various *C. oleifera* cultivars with different inoculation methods, the growth of *C. oleifera* cultivars could be obviously increased, and the antagonistic endophytic bacterium CSUFT-BV4 could significantly increase the activities of various defense enzymes in *C. oleifera* and induced resistance to *C. oleifera* anthracnose. The growth-promoting and antagonistic activity of CSUFT-BV4 implied that this strain can be regarded as a biological control agent to promote the growth of *C. oleifera* plants and to control anthracnose diseases in future production.

## Figures and Tables

**Figure 1 microorganisms-12-00763-f001:**
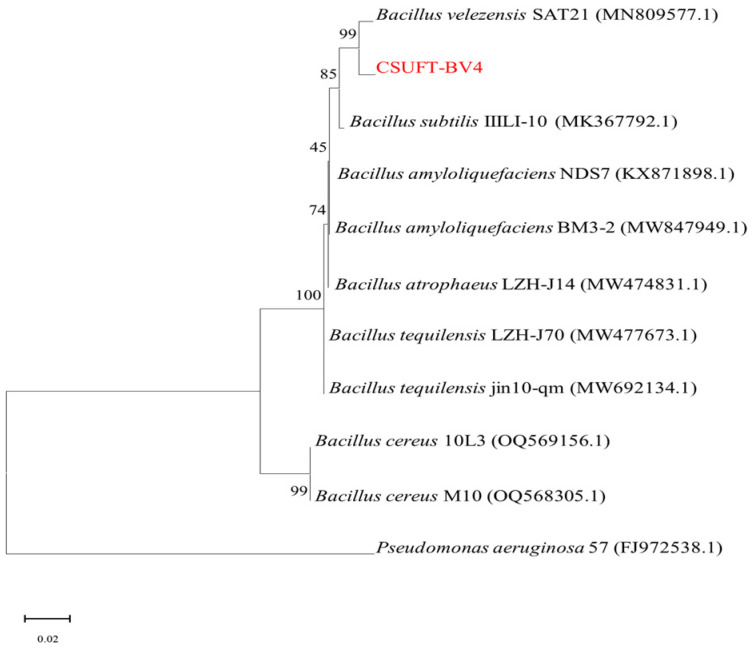
Phylogenetic tree of CSUFT-BV4 based on 16S rDNA sequence.

**Figure 2 microorganisms-12-00763-f002:**
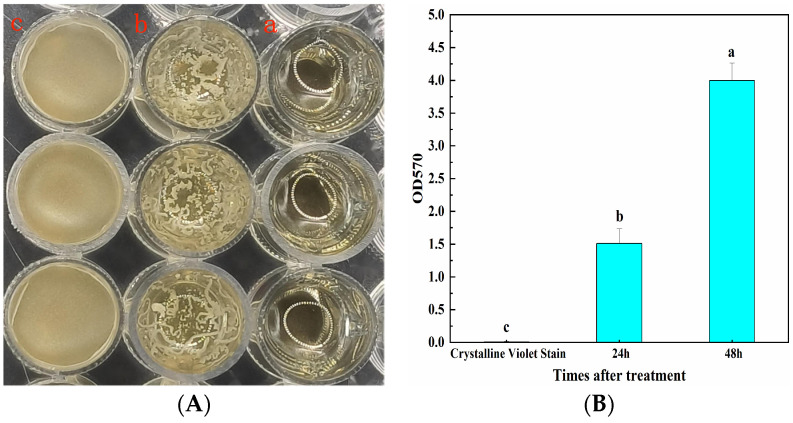
The biofilm formation of CSUFT-BV4. (**A**): Biofilm formed on the surface (a: liquid LB control; b: bacterial liquid culture for 24 h; c: bacterial liquid culture for 48 h); (**B**) quantification of CSUFT-BV4 biofilm. Error bars represent standard deviations. According to Duncan’s test, different lowercase letters above each column represent significant differences between treatments (*p* < 0.05).

**Figure 3 microorganisms-12-00763-f003:**
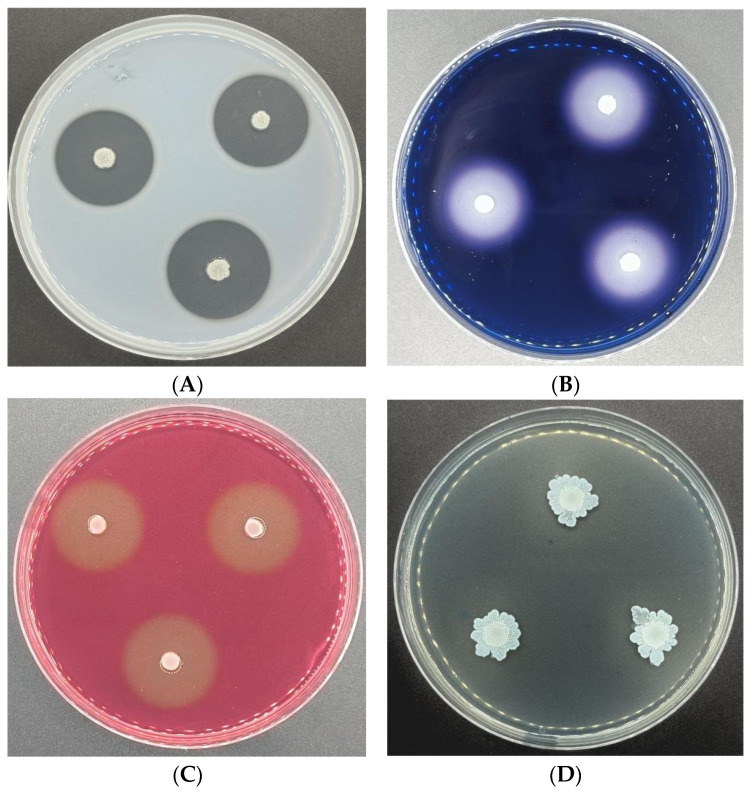
The ability of CSUFT-BV4 to produce protease (**A**), amylase (**B**), cellulase (**C**), and β-1,3-glucanase (**D**).

**Figure 4 microorganisms-12-00763-f004:**
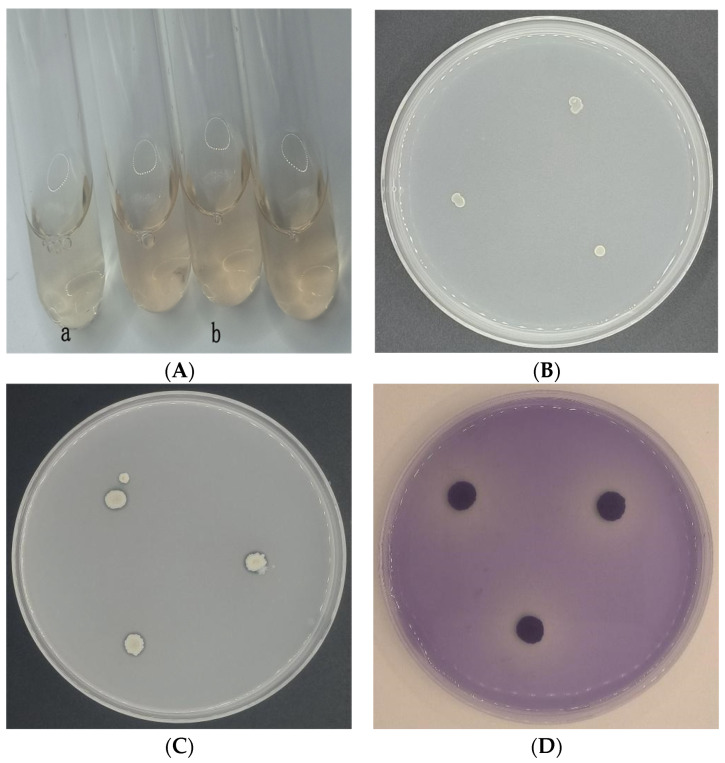
The growth-promoting characteristics of CSUFT-BV4. (**A**) Producing indoleacetic acid, a: CK LB control, b: CSUFT-BV4 fermentation broth; (**B**) nitrogen fixation; (**C**) organic phosphorus testing plate; (**D**) inorganic phosphorus testing plate.

**Figure 5 microorganisms-12-00763-f005:**
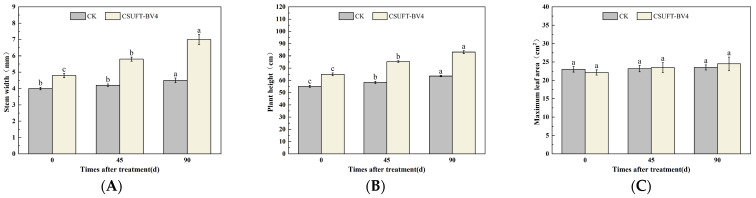
Effect on growth of “Huashuo” *C. oleifera* cultivars by leaf spray method. (**A**) Stem width; (**B**) plant height; (**C**) maximum leaf area. CK: LB control; CSUFT-BV4: 10^8^ CFU/mL fermentation broth. Error bars indicate standard deviation. Different lowercase letters above each column represent significant differences according to Duncan’s test (*p* < 0.05).

**Figure 6 microorganisms-12-00763-f006:**
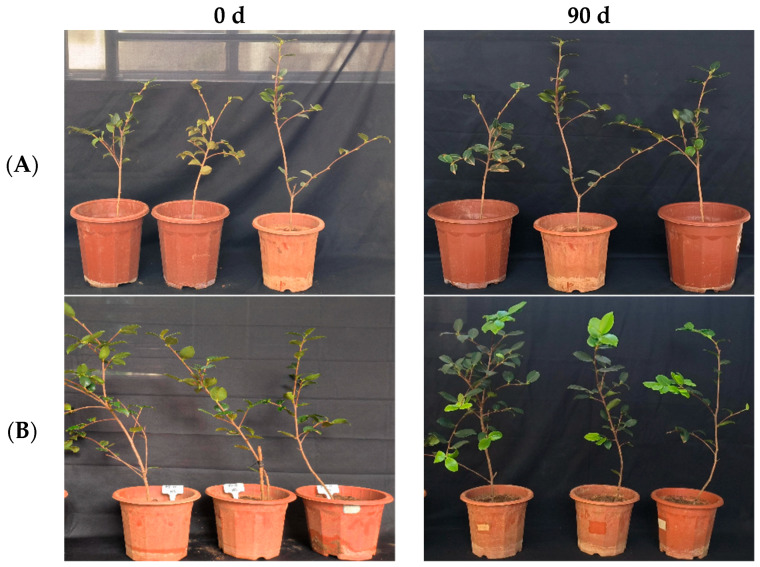
Growth changes in “Huashuo” *C. oleifera* cultivars by leaf spray method. (**A**) CK LB control; (**B**) inoculated with 10^8^ CFU/mL CSUFT-BV4 fermentation broth.

**Figure 7 microorganisms-12-00763-f007:**
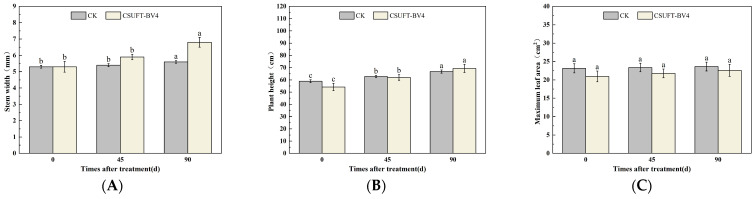
Effect on growth of “Huashuo” *C. oleifera* cultivars by root irrigation method. (**A**) Stem width; (**B**) plant height; (**C**) maximum leaf area. CK: LB control; CSUFT-BV4: 10^8^ CFU/mL fermentation broth. Error bars indicate standard deviation. Different lowercase letters above each column represent significant differences according to Duncan’s test (*p* < 0.05).

**Figure 8 microorganisms-12-00763-f008:**
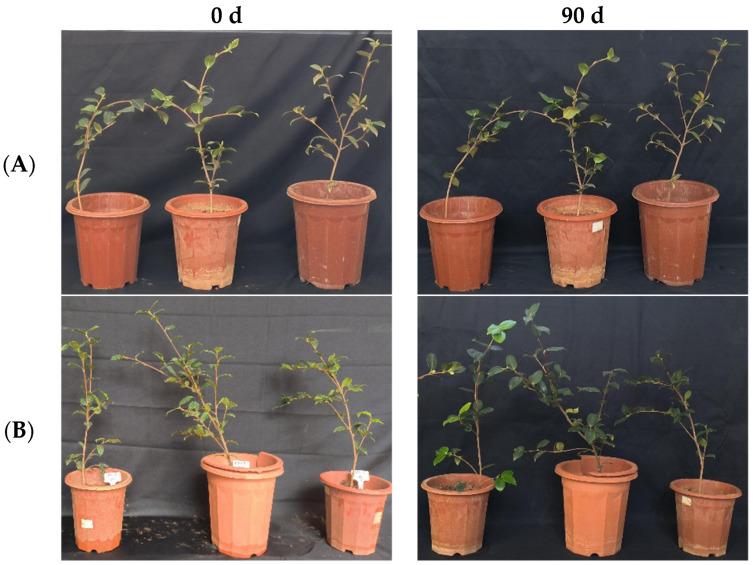
Growth changes in “Huashuo” *C. oleifera* cultivars by root irrigation method. (**A**) CK LB control; (**B**) inoculated with 10^8^ CFU/mL CSUFT-BV4 fermentation broth.

**Figure 9 microorganisms-12-00763-f009:**
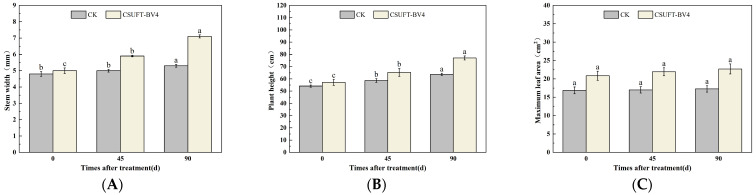
Effect on growth of “Huashuo” *C. oleifera* cultivars by leaf spray + root irrigation method. (**A**) Stem width; (**B**) plant height; (**C**) maximum leaf area. CK: LB control; CSUFT-BV4: 10^8^ CFU/mL fermentation broth. Error bars indicate standard deviation. Different lowercase letters above each column represent significant differences according to Duncan’s test (*p* < 0.05).

**Figure 10 microorganisms-12-00763-f010:**
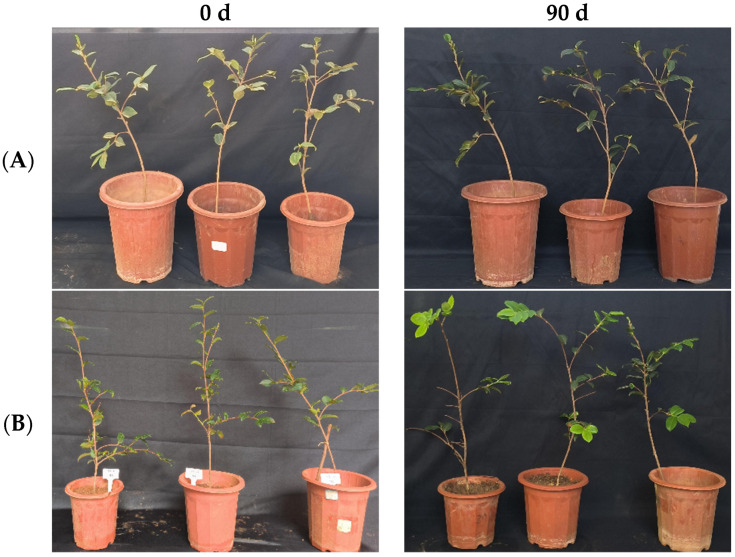
Growth changes in “Huashuo” *C. oleifera* cultivars by leaf spray + root irrigation method. (**A**) CK LB control; (**B**) inoculated with 10^8^ CFU/mL CSUFT-BV4 fermentation broth.

**Figure 11 microorganisms-12-00763-f011:**
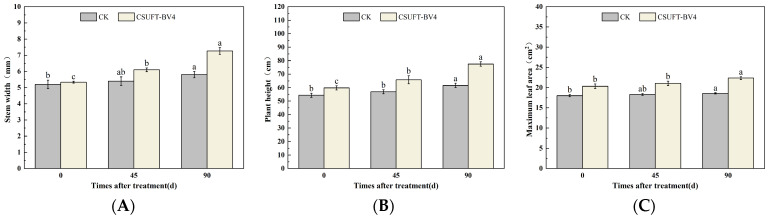
Effect on growth of “Huajin” *C. oleifera* cultivars by leaf spray method. (**A**) Stem width; (**B**) plant height; (**C**) maximum leaf area. CK: LB control; CSUFT-BV4: 10^8^ CFU/mL fermentation broth. Error bars indicate standard deviation. Different lowercase letters above each column represent significant differences according to Duncan’s test (*p* < 0.05).

**Figure 12 microorganisms-12-00763-f012:**
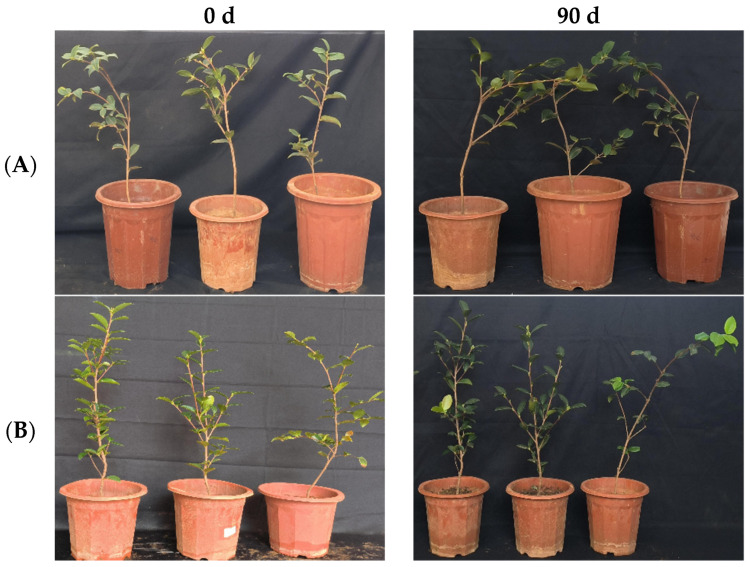
Growth changes in “Huajin” *C. oleifera* cultivars by leaf spray method. (**A**) CK LB control; (**B**) inoculated with 10^8^ CFU/mL CSUFT-BV4 fermentation broth.

**Figure 13 microorganisms-12-00763-f013:**
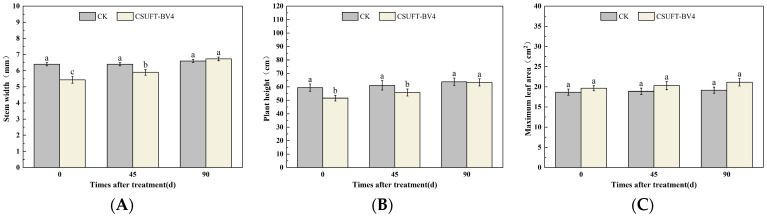
Effect on growth of “Huajin” *C. oleifera* cultivars by root irrigation method. (**A**) Stem width; (**B**) plant height; (**C**) maximum leaf area. CK: LB control; CSUFT-BV4: 10^8^ CFU/mL fermentation broth. Error bars indicate standard deviation. Different lowercase letters above each column represent significant differences according to Duncan’s test (*p* < 0.05).

**Figure 14 microorganisms-12-00763-f014:**
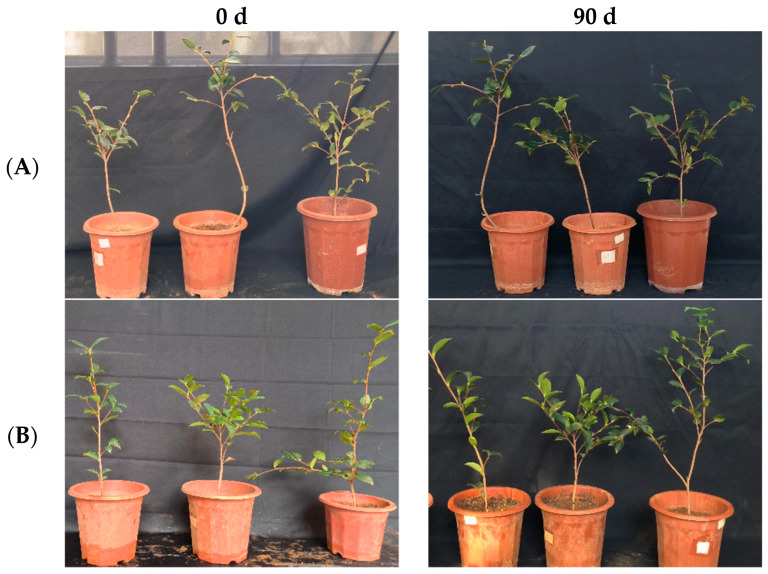
Growth changes in “Huajin” *C. oleifera* cultivars by root irrigation method. (**A**) CK LB control; (**B**) inoculated with 10^8^ CFU/mL CSUFT-BV4 fermentation broth.

**Figure 15 microorganisms-12-00763-f015:**
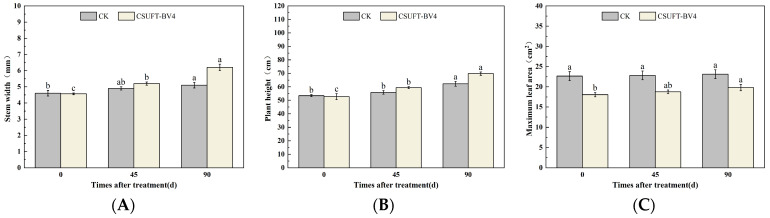
Effect on growth of “Huajin” *C. oleifera* cultivars by leaf spray + root irrigation method. (**A**) Stem width; (**B**) plant height; (**C**) maximum leaf area. CK: LB control; CSUFT-BV4: 10^8^ CFU/mL fermentation broth. Error bars indicate standard deviation. Different lowercase letters above each column represent significant differences according to Duncan’s test (*p* < 0.05).

**Figure 16 microorganisms-12-00763-f016:**
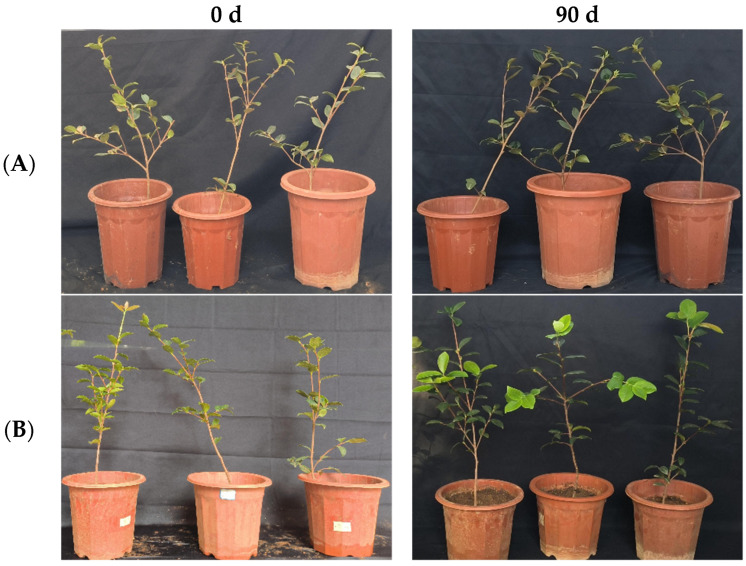
Growth changes in “Huajin” C. oleifera cultivars by leaf spray + root irrigation method. (**A**) CK LB control; (**B**) inoculated with 10^8^ CFU/mL CSUFT-BV4 fermentation broth.

**Figure 17 microorganisms-12-00763-f017:**
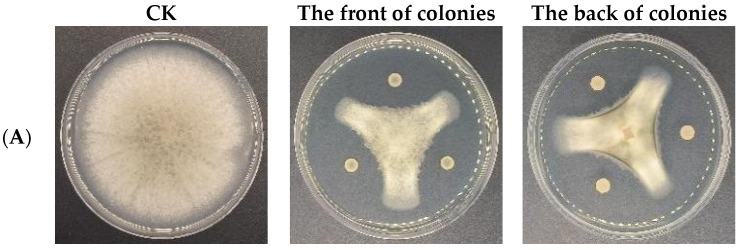
Antagonistic activities of CSUFT-BV4 against *C. oleifera* anthracnose in vitro. (**A**): *C. fructicola*; (**B**): *C. gloeosporioides*; (**C**): *C. siamense*; (**D**). *C. camelliae*; (**E**): *C. kahawae*.

**Figure 18 microorganisms-12-00763-f018:**
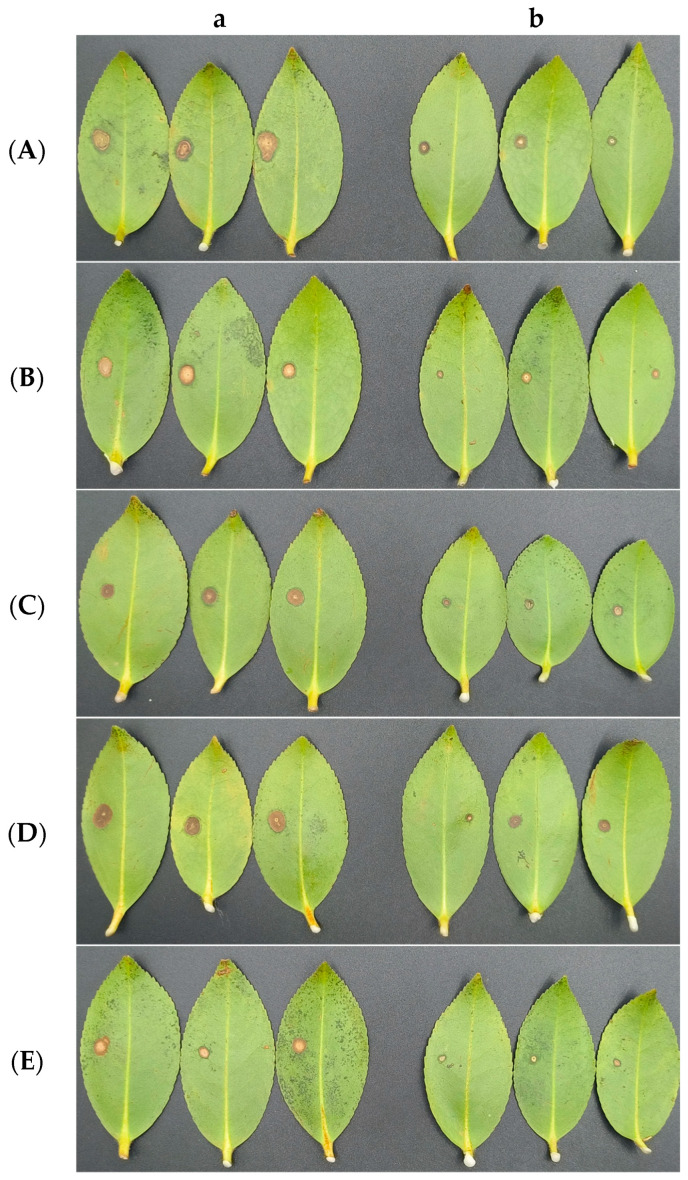
Antagonistic activities of CSUFT-BV4 against *C. oleifera* anthracnose in vivo. (**A**): *C. fructicola*; (**B**): *C. gloeosporioides*; (**C**): *C. siamense*; (**D**): *C. camelliae*; (**E**): *C. kahawae*. a: CK, control, inoculated with pathogens alone; b: inoculated with CSUFT-BV4 fermentation broth.

**Figure 19 microorganisms-12-00763-f019:**
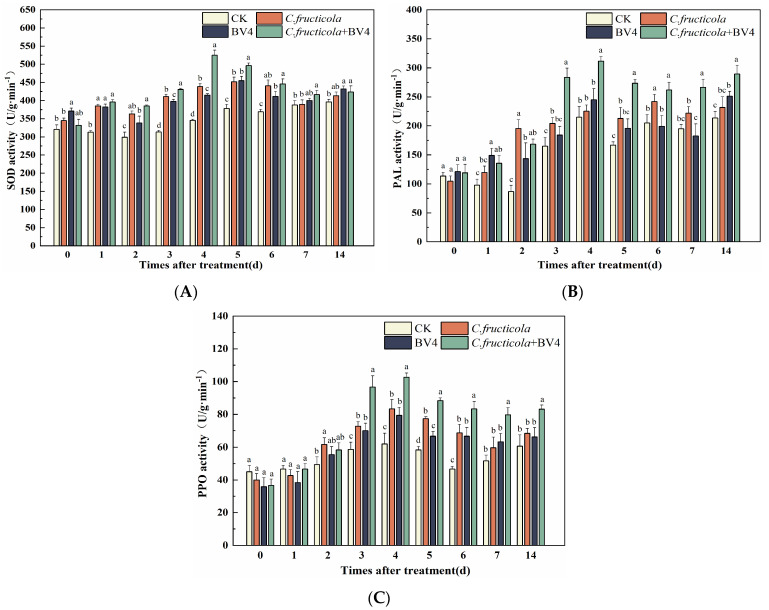
Changes in defense enzyme activities in *C. oleifera*. (**A**) Superoxide dismutase SOD; (**B**) phenylalanine aminotransferase PAL; (**C**) polyphenol oxidase PPO. CK: control, inoculated with sterilized LB liquid medium alone; *C. fructicola*: inoculated with spore solution of pathogen alone; CSUFT-BV4: inoculated with the fermentation broth alone; *C.fructicola* + CSUFT-BV4: inoculated with both the pathogen *C. fructicola* and the endophytic bacterium CSUFT-BV4. Error bars indicate standard deviation. Different lowercase letters represent significant differences between treatments at the same time (*p* < 0.05).

**Table 1 microorganisms-12-00763-t001:** The metabolic substances synthesized by CSUFT-BV4.

Metabolic Substances	CSUFT-BV4
Protease	+
Amylase	+
Cellulase	+
β-1,3-glucanase	+

“+” represents a positive result for the ability to synthesize metabolites.

**Table 2 microorganisms-12-00763-t002:** The growth-promoting characteristics of CSUFT-BV4.

Growth-Promoting Characteristics	CSUFT-BV4
Indoleacetic acid production	+
Nitrogen fixation	+
Organic phosphorus solubilizing	+
Inorganic phosphorus solubilizing	+
Potassium solubilizing	−
Siderophore secretion	−

“+” represents a positive result and “−” represents a negative result.

**Table 3 microorganisms-12-00763-t003:** The inhibition rate of CSUFT-BV4 against *C. oleifera* anthracnose pathogens in vitro.

Strain	Inhibition Rate (%)			
	Treatment 1	Treatment 2	Treatment 3	Average
CK				
*C. fructicola*	74.2	72.8	72.6	73.2 ± 0.71 ^a^
*C. gloeosporioides*	70.7	72.6	71.3	71.5 ± 0.82 ^ab^
*C. siamense*	70.0	68.1	68.2	68.8 ± 0.89 ^b^
*C. camelliae*	71.3	72.0	66.4	69.9 ± 2.49 ^ab^
*C. kahawae*	72.5	71.9	69.6	71.4 ± 1.26 ^ab^

Different lowercase letters at the tail of each column represent significant differences according to Duncan’s test (*p* < 0.05).

**Table 4 microorganisms-12-00763-t004:** The inhibition rate of CSUFT-BV4 against anthracnose pathogens in vivo.

Strain	Inhibition Rate (%)			
	Treatment 1	Treatment 2	Treatment 3	Average
CK				0
*C. fructicola*	58.8%	63.3%	62.5%	61.6% ± 1.95 ^a^
*C. gloeosporioides*	60.0%	62.5%	60.0%	60.8% ± 1.18 ^a^
*C. siamense*	62.0%	54.0%	58.3%	58.1% ± 3.27 ^a^
*C. camelliae*	62.5%	50.8%	52.9%	55.4% ± 5.1 ^a^
*C. kahawae*	61.0%	62.0%	55.6%	59.5% ± 2.8 ^a^

Same lowercase letters at the top of each column represent non-significant differences according to Duncan’s test (*p* < 0.05).

## Data Availability

All relevant data are within this paper.

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
