# Peer review of "The Research of Antagonistic Endophytic Bacterium Bacillus velezensis CSUFT-BV4 for Growth Promotion and Induction of Resistance to Anthracnose in Camellia oleifera"

_microorganisms, 2024, doi:10.3390/microorganisms12040763_

Round 1
Reviewer 1 Report
Comments and Suggestions for Authors
Review of manuscript ID microorganisms-2964283 and titled "The research of antagonistic endophytic bacterium CSUFT-BV4 for growth promotion and induction of resistance to anthracnose in Camellia oleifera”. I believe that the work concerns an important phytopathological aspect from a scientific and application perspective. I did not notice any serious technical or methodological shortcomings in the work. The manuscript is written in a clear and coherent way. The results obtained are interesting and well documented. Therefore, I recommend the manuscript for publication in the Microorganisms journal.
Reviewer 2 Report
Comments and Suggestions for Authors
I congratulate the authors for their work and manuscript. Manuscript ID: microorganisms-2964283 "The research of antagonistic endophytic bacterium CSUFT-BV4 for growth promotion and induction of resistance to anthracnose in Camellia oleifera" by He and collaborators describes the investigation of the potential to control a destructive disease of this Chinese plant. It is an important source of oil with growing importance within China, but the manuscript should be interesting to the broad readership of Microorganisms. The authors were able to identify the potential biocontrol strain of Bacillus using 16S sequencing, demonstrate a series of enzymatic activities, show its plant-growth promoting capability, and finally its inhibitory effect on the growth of anthracnose pathogens, both in vitro and in planta. An increase in defense enzymes following treatment with the biocontrol was also demonstrated, reinforcing the notion of the biocontrol "priming" the plant immune system against anthracnose pathogens.
Overall the manuscript is well written, with methods clearly described and results clearly presented. The conclusions are supported by the results and I don't see any major issues. I recommend acceptance in the current format.
Figure 2A: Consider replacing the red-colored letters with white circles containing black letters for visual clarity.
Although the manuscript is objective and clear, some minor improvements will be necessary to reach the quality standards of Microorganisms.